# Prevalence and Genetic Diversity of Rotaviruses among under-Five Children in Ethiopia: A Systematic Review and Meta-Analysis

**DOI:** 10.3390/v12010062

**Published:** 2020-01-03

**Authors:** Debasu Damtie, Mulugeta Melku, Belay Tessema, Anastasia N. Vlasova

**Affiliations:** 1Department of Immunology and Molecular Biology, School of Biomedical and Laboratory Sciences, College of Medicine and Health Sciences, University of Gondar, Gondar, Ethiopia; 2Food Animal Health Research Program, CFAES, Ohio Agricultural Research and Development Center, Department of Veterinary Preventive Medicine, The Ohio State University, Wooster, OH 44691, USA; 3Department of Hematology and Immunohematology, School of Biomedical and Laboratory Sciences, College of Medicine and Health Sciences, University of Gondar, Gondar, Ethiopia; mulugeta.melku@gmail.com; 4Department of Medical Microbiology, School of Biomedical and Laboratory Sciences, College of Medicine and Health Sciences, University of Gondar, Gondar, Ethiopia; bt1488@yahoo.com

**Keywords:** acute gastroenteritis, diarrhea, genotype diversity, rotavirus infection, meta-analysis, Ethiopia

## Abstract

Rotavirus infection is the major cause of acute gastroenteritis among children globally. Sub-Saharan Africa including Ethiopia is disproportionally affected by the disease. The aims of this review were to determine the pooled prevalence of rotavirus infection among children under-five and to identify the dominant rotavirus genotypes in Ethiopia. Twelve studies were included to estimate the pooled prevalence of rotavirus acute gastroenteritis and five studies were used to determine predominantly circulating genotypes of rotavirus. The pooled prevalence of rotavirus infection was 23% (95% CI = 22%–24%). G3 (27.1%) and P[8] (49%) were the dominant G and P types, respectively. The G8 G-type uncommon in humans but highly prevalent in cattle was also reported accounting for 1% of all cases. The major G/P combinations were G12P[8] (15.4%), G3P[6] (14.2%), G1P[8] (13.6%) and G3P[8] (12.9%) collectively accounting for 56.1% of rotavirus strains. Similar to other parts of the world, the dominance of G1, G3, P[6] and P[8] genotypes was noted in Ethiopia. The increased prevalence of G12P[8] strains observed in Ethiopia was similar to observations in other geographic regions in the post-vaccine introduction period. Thus, further studies are required on the vaccine effectiveness, genotype distribution and inter-species transmission potential of rotaviruses in Ethiopia.

## 1. Introduction

Diarrhea is the second most frequent cause of death among young children globally following respiratory tract infections [1]. In developing countries, diarrhea is responsible for 25%–30% of deaths [2]. Rotavirus remains the leading cause of severe gastroenteritis worldwide [3]. According to the World Health Organization report in 2016, there were 215,000 deaths due to rotavirus infection in children less than five years old in 2013 globally. Of these, Sub-Saharan Africa accounted for 121,000 (56.3%) deaths. In Ethiopia there were 6817 deaths due to rotavirus infection in the same year [4].

Rotaviruses are triple-layered particles containing 11 segments of genomic double-stranded RNA that belong to the *Reoviridae* family. The virus encodes six structural proteins (VP1–VP4, VP6 and VP7) and six non-structural proteins (NSP1–NSP6) [5]. Rotaviruses are classified into different P and G-types based on the genetic characteristics of their VP4 and VP7 genes respectively. Studies have shown wide spatio-temporal variations in G- and P-type distribution and prevalence on different continents [6,7]. The enormous diversity of rotaviruses is achieved via point mutations, recombination and gene reassortment [8]. Therefore, continuous rotavirus surveillance is needed to monitor the prevalence and possible changes of the dominant G and P types circulating in a given region over time.

The predominant rotavirus A genotypes are G1, G2, G3, G4 and G9 in conjunction with P[8], P[6] and P[4] and represent over 90% of the human strains worldwide [9]. Since its first isolation in children in the Philippines in 1987, the G12 genotype has emerged through reassortment with local strains and spread globally [10,11]. Increased prevalence of the G12P[8] genotype combination was reported for the first time in Brazil in the post-vaccine introduction period [12]. However, there is high genetic diversity of rotavirus strains circulating in Africa [13]. This G–P difference could be one of the reasons for variable efficacy of the available vaccines in different geographic locations [14]. Identification of the predominantly circulating genotypes of the rotavirus is essential to achieve optimal vaccine performance. A monovalent G1P[8] rotavirus vaccine (Rotarix) was introduced and included into the Expanded Program on Immunization (EPI) program in Ethiopia in 2013 [15]. Up to date, no studies to evaluate the performance of rotavirus vaccines in the region have been done. Also, there have been very few and incomplete studies conducted on the magnitude and even much less data on genetic distribution of rotavirus infection among children in Ethiopia. The objective of this systematic review and meta-analysis was, therefore, to determine pooled rates of rotavirus infection at the national level and the genotype distribution of rotavirus in Ethiopia based on the available data sources.

## 2. Review Questions

The research questions were developed based on the condition, context and population (CoCoPo) approach. There were two questions that this review aimed to answer:Is the pooled prevalence of rotavirus infection among under-five children with acute gastroenteritis different in Ethiopia compared to other African countries and geographic regions?Are dominant rotavirus genotypes currently circulating in Ethiopia different from those in other countries or enriched for strains of zoonotic origin?

## 3. Methods

### 3.1. Review Protocol Development

The PROSPERO and Cochran data bases were checked for the presence of similar studies to avoid duplication. However, we found no similar studies registered in the aforementioned libraries. With the objective of answering our review questions, the protocol was developed and registered on PROSPERO with the registration number of CRD42019129578.

### 3.2. Search Strategy

We searched primary studies on the prevalence and genetic distribution of rotavirus infection among under-five years of age children from PubMed, Embase, ScienceDirect, African Journals Online and Google Scholar databases to identify studies published in English. Boolean operators (not, and, or) were also used in succession to narrow and widen the scope of the search. Publications were identified using the search terms “epidemiology”, “prevalence”, “burden”, “rotavirus”, “rotavirus infection”, “acute gastroenteritis”, “acute diarrhea”, “Ethiopia” and related terms. Other articles were identified by reviewing the reference list of the primary articles. All the searched articles were imported to Endnote version 7 bibliography management tool to avoid duplicates and manage the citations.

### 3.3. Selection Criteria of Studies

Inclusion criteria: The primary selection was made based on the major topic of the article. In addition, the following pre-defined inclusion criteria were used:The study participants must be children with acute diarrhea/gastroenteritis;The age of the study participants must be under-five years of age andRotavirus positive and negative results must be reported.

Exclusion criteria: Studies were excluded if they did not include the required information indicated in the inclusion criteria.

### 3.4. Study Selection and Quality Assessment

Full-length original research articles were obtained and assessed in detail by two independent reviewers using the Joanna Briggs Institute (JBI) Critical Appraisal Checklist for Prevalence Studies. Before the final decision to include or exclude the articles, the independent reviewers sat together and settled the differences in rating through consensus. Articles with average score of 50% and above were included into this study. To assess the proportion of rotavirus infection among children, we included all observational studies (cross-sectional and surveillance) that included children under five years of age with symptoms of acute gastroenteritis that had used enzyme immune assay (EIA) or enzyme linked immunosorbent assay (ELISA) and reverse transcription polymerase chain reaction (RT-PCR) for the identification of rotavirus. The studies were disaggregated as pre-vaccination and post-vaccination periods. The rotavirus vaccine was introduced in Ethiopia in 2013. Therefore, all studies conducted before 2013 were considered as pre-vaccination studies; while studies after the indicated year were considered as post-vaccination period studies. All studies were included in the description of genotypes if they had used reverse transcription polymerase chain reaction (RT-PCR) to identify the circulating strains. For the description of strain distribution, we included studies reporting the number of samples tested and the G and P combinations.

### 3.5. Data Extraction

The table for data extraction was developed in Microsoft Excel. The information extracted included the author’s name, publication year, region, study period, study design, sample size, number of rotavirus-positive children, study setting (facility based or community based), proportion of rotavirus cases, genotypes identified and respective frequencies of identified genotypes. Not all studies reported rotavirus genotype distribution. Hence, the weighted proportions of each genotype of rotavirus were calculated by dividing the frequency of a particular strain of rotavirus to total isolates reported by the primary studies.

### 3.6. Statistical Analysis

The extracted data was imported from Excel to STATA version 14. The pooled proportion of rotavirus infection among children with acute gastroenteritis was determined by using a fixed effect model as there was no heterogeneity between primary studies. The weighted proportion of each genotype of rotavirus was determined by dividing the number of specific genotype detections by the total number of genotyped samples in Microsoft Excel. The heterogeneity test was done using both subjective (forest plot and Galbraith plot) and objective methods (I^2^ test and Cochran’s Q-test). Heterogeneity among studies was considered significant if the *p* value of Cochran’s Q-test is less or equal to 0.05. The full texts of the primary studies and the extracted data in a spreadsheet are available from the authors.

## 4. Results

### 4.1. Characteristics of Individual Studies

We have accessed a total of 3961 research articles from PubMed, Embase, ScienceDirect, African Journals Online and other sources after duplicates were removed. Of these, 3948 were removed after reading the titles and abstracts of the articles; the reasons being the titles did not match our interest (3854 articles), some articles lacked title and abstract (33 articles) and abstract did not contain required data (61 articles). Full texts of the remaining 13 research articles were critically reviewed and assessed for their quality using the JBI quality appraisal checklist for prevalence studies (Figure 1). Eleven out of 13 full-length articles were included in the systematic review and meta-analysis. Two of the articles were excluded after reading the full-length article; one article was excluded because of reporting the cases without denominators and the other presented the data inappropriately (Figure 1). One of the research articles included for analysis contained information on the prevalence of childhood rotavirus infections in pre- and post-vaccination periods. As a result, it was treated as two separate studies considering the pre-vaccination and post-vaccination data as a separate article. The rest of the studies reported either the pre-vaccination or the post-vaccination data. Finally, we used 12 articles to estimate the pooled prevalence of rotavirus infection among under-five years of age children in Ethiopia. Among the 12 articles only five reported on the genotype distribution of rotavirus infection.

The majority of the research articles were from Addis Ababa City Administration (7/12); cross-sectional studies by design (8/12), conducted before the introduction of the rotavirus vaccine in Ethiopia (8/12) and almost all studies (11/12) used the EIA method for the determination of rotavirus infection from stool specimens of diarrheic children. All the five studies that reported genotype data used RT-PCR as the laboratory method of analysis (Table 1).

### 4.2. Prevalence of Rotavirus Infection among under-Five Children in Ethiopia

The 12 studies analyzed 6509 stool samples collected from under-five children with acute gastroenteritis to determine the proportion of rotavirus infection. Of these, 1580 were found to be rotavirus positive. Pooled rotavirus infection prevalence among children under-five was determined from the available data. Heterogeneity refers to the variation in study outcomes between studies included for meta-analysis. A heterogeneity test in Galbraith plot revealed that the estimates of all studies lay between the ±2 standard error which is an indication of the absence of significant heterogeneity (Figure 2).

The forest plot and objective assessment of the data (I^2^ = 0.00%) also supported the absence of substantial heterogeneity between individual studies. We also ran a funnel plot test to evaluate whether there was publication bias. A visual inspection of the resultant funnel plot revealed asymmetrical distribution of the study findings (Figure 3). However, the objective assessment of bias using the Egger’s regression test (Figure 4, Table 2) revealed the absence of publication bias (*p* = 0.619).

The prevalence of rotavirus infection among children with acute gastroenteritis in Ethiopia ranged from 8.04% in a community-based study in Amhara regional state to 43.63% in a health facility-based study in Oromia regional state. The pooled prevalence of rotavirus infection among under-five children with acute gastroenteritis was 23% (95% CI = 22%, 24%) (Figure 5).

We have also looked at the prevalence of rotavirus infection before and after the introduction of rotavirus in Ethiopia. The result showed no significant difference in the prevalence of rotavirus infection between the pre-vaccine introduction 24% (95% CI: 23, 25) and post-vaccine introduction 21% (95% CI = 19, 23) periods (Figure 6 and Figure 7).

### 4.3. Genotype Diversity of Rotavirus in Ethiopia

Information about the G and P type of rotavirus A was available for 719 strains from five studies (Figure 8A,B). Most of the strains were reported from Addis Ababa which accounted for 564 (78.44%). The genotypic data from the five studies revealed that the circulating G-types included G1, G2, G3, G8, G9 and G12. The circulating P-types included P[4], P[6], and P[8]. Mixed G and P types were also reported from diarrheic children.

G3 was the most dominant G type (27.1%) followed by G12 (17.2%) and G1 (17%). The most common P types were P[8] (49%), P[6] (22.4%) and P[4] (12.3%).

The major G-types identified before the introduction of the vaccine in Ethiopia were G1 (24%), G12 (21%), G3 (17%) and G2 (12%) while G3 (45%), G2 (14%), G9 (13%) and G12 (11%) were the commonly identified genotypes post-vaccine introduction (Figure 9). The decrease in G1 and an increase in other G-types can be related to the selective pressure of the vaccine on G1.

This review study also demonstrated P[8] (43%), P[6] (29%) and P[4] (11%) were the commonly identified P-types of rotavirus before the introduction of a rotavirus vaccine. Likewise the P[8] (60%), P[6] (11%) and P[4] (15%) remained the commonly identified P-types post-vaccine introduction with P[8] becoming more dominant than before (Figure 10).

The four major G/P combinations in Ethiopia were G12P[8] (15.43%), G3P[6] (14.2%), G1P[8] (13.6%) and G3P[8] (12.9%) (Table 3). Before the introduction of a rotavirus vaccine, G1P[8] (19.78%), G12P[8] (18.49), G3P[6] (16.77%) and G4P[4] (20.45%) were the leading G/P combinations. However, G3P[8] (35.83%), G2P[4] (14.17%), G9P[8] (12.2%) and G12P[8] (9.94%) became dominant G/P combinations post-rotavirus vaccine introduction in Ethiopia. Which implies that the introduction of the vaccine could potentially affect the distribution of major rotavirus genotypes.

## 5. Discussion

Rotavirus infection is a significant public health concern in developing countries with poor socio-economic situations and the lack of appropriate sanitation and hygiene. This meta-analysis estimated the pooled prevalence of rotavirus infection among under-five children with acute gastroenteritis in Ethiopia to be 23% (95% CI = 22%, 24%). This finding indicates that a considerable proportion of acute gastroenteritis among under-five children in Ethiopia is caused by rotavirus A infection. This finding is consistent with a meta-analysis report from Latin America and Caribbean countries which reported a pooled prevalence of 24.3% [27]. This can potentially lead to mismanagement of rotavirus acute gastroenteritis with antibiotics in the absence of appropriate diagnostic tools for acute viral gastroenteritis in health facilities of Ethiopia; which in turn contributes to the problem of antimicrobial resistance in the country. However, this prevalence is lower than that in a multinational and multicenter African rotavirus surveillance network report from eight African countries which reported an overall prevalence of rotavirus infection of 40% [26]. Similarly, two systematic reviews and meta-analysis studies in Iranian children reported a pooled prevalence of 35% and 39.9% [28,29]. The variations could be explained by the difference in the study subjects/inclusion criteria; for example, the African rotavirus surveillance included only severely ill hospitalized diarrheic children while in our meta-analysis all children with acute diarrhea were included regardless of the severity of the disease which may in turn underestimate the prevalence. In addition to study subject selection criteria, the study period may have contributed to the high prevalence for those studies which were conducted before the introduction of a rotavirus vaccine. The studies included in our meta-analysis were conducted both pre- and post-vaccine introduction periods which may lower the prevalence.

Rotavirus genotypes varied over time and the peak frequency of one strain was often followed by a replacement by a different genotype [9]. In this meta-analysis the most common G type was G3 (27.1%). This finding is similar to another systematic review and meta-analysis study from China that identified G3 (39.3%) as the most common G type [30]. Unlike our results, other studies in Africa as a continent and in sub-Saharan Africa as a region reported G1 as the dominant G-type with 32.72% and 34.9% frequency respectively [31,32]. Likewise, another study in China reported G1 (39.5%) followed by G3 (35.6%) as the most dominant G types [33]. An unusual G-type in humans (G8) was also reported which is common in cattle. This indicates that humans and animals could have common strains that can be transmitted between populations of different species. Generally, the G types of rotavirus vary by geography and unusual strains may emerge as it is evident in the reports. Therefore, periodic surveys of local rotavirus strains to identify the predominant genotypes circulating is crucial to target strain specific interventions such as vaccination.

The dominant P type in this study was P[8] (49%). This finding is consistent with reports from two studies in China of P[8] (50.2%) and P[8] (54.6%) [30,33]. Several other studies identified P[8] as a predominant P-type in Africa P[8] (48.71%) and in sub-Saharan Africa P[8] (35.5%) [31,32]. Unlike the G types, P types of rotavirus tend to be less variable across different geographic locations.

On the other hand, the dominant G/P combinations in this study were G12P[8], G3P[6], G1P[8] and G3P[8] which accounted for 15.43%, 14.2%, 13.6% and 12.9% of the rotavirus strains respectively. The four major G/P combinations accounted over 55% of rotavirus strains among children with acute gastroenteritis in Ethiopia. The observation that G1, G3, G12, P[6] and P[8] genotypes were predominant is consistent with a six-year rotavirus surveillance data in South and East African countries which also demonstrated that 23.8% of the strains were G1P[8], followed by G2P[4] (11.8%), G9P[8](10.4%), G12P[8] (4.9%), G2P[6] (4.2%) and G3P[6] (3.7%) [34]. Another meta-analysis study done in Africa has also reported G1P[8] (22.64%), G2P[4] (8.29%), G9P[8] (6.95%) and G2P[6] (5%) as the four dominant G/P combinations [32]. Likewise, our finding is in line with the African rotavirus surveillance network report showing G1P[8] (21%) and G2P[4] (7%) as the two predominant G/P combinations [26]. A study in China reported G3P[8] (32.1%), G1P[8] (23.0%) and G2P[4] (7.9%) as the most common G/P combinations [30]. Another study in China also reported G3P[8] (32.1%) and G1P[8] (24.5%), followed by G2P[6] (13.2%) and G2P[4] (10.1%) as the most common G/P combinations [33]. Although, the most commonly reported G/P combinations from different geographical regions seem similar, the proportion accounted by each rotavirus G/P combination vary per geographic region. In contrast to most previous reports, our analysis identified G12P[8] as the dominant genotype constellation. The G12P[8] strain was the leading genotype combination circulating in Ethiopia. This strain showed an increasing trend when we compared it with a previous African report of 3.1% [13]. The increasing prevalence of G12P[8] and other unusual G8P[6] and G8P[8] genotypes could be related with their stability as there is limited heard immunity for newly emerging strains. It is important to notice that the prevalence of G12P[8] rotavirus strains has increased globally [12,35,36,37] in the post-vaccine introduction period, suggesting that they could have emerged as escape variants. The currently available vaccine in Ethiopia is a monovalent Rotarix vaccine that covers the G1P[8] strain; however, it may or may not provide sufficient protections against heterologous and the newly emerging rotavirus strains [14]. Hence, the development of new strain specific rotavirus vaccines for different regions based on surveillance data is crucial.

## 6. Limitations of the Review Study

In this review study, only articles published in English were used for our literature search. Moreover, because of the paucity of primary studies, very few primary studies were included to pool the genotype diversity of rotavirus infection in Ethiopia.

## 7. Conclusions and Recommendations

The prevalence of rotavirus acute gastroenteritis among children in Ethiopia is considerable which accounted for nearly a quarter of all acute gastroenteritis cases. The genotype distribution of rotavirus infection in Ethiopia is somewhat similar to other parts of Africa and global strain distribution. However, there is a relative increase in the distribution of emerging unusual rotavirus strains of animal origin in Ethiopia including G12 and G8. Therefore, continuous surveillance of rotavirus infection to identify the predominantly circulating strain in Ethiopia has to be considered to design and implement appropriate strain specific interventions including new strain specific vaccine development. Moreover, studies are recommended on vaccine effectiveness, genotypic diversity and zoonotic transmission potential of rotavirus infection.

## Figures and Tables

**Figure 1 viruses-12-00062-f001:**
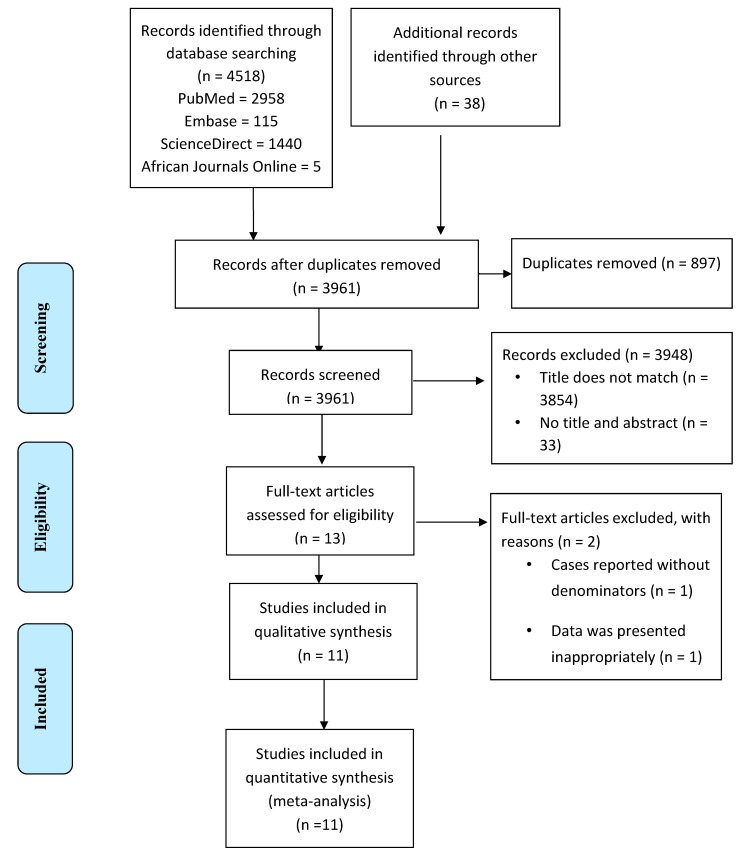
Study selection flow diagram.

**Figure 2 viruses-12-00062-f002:**
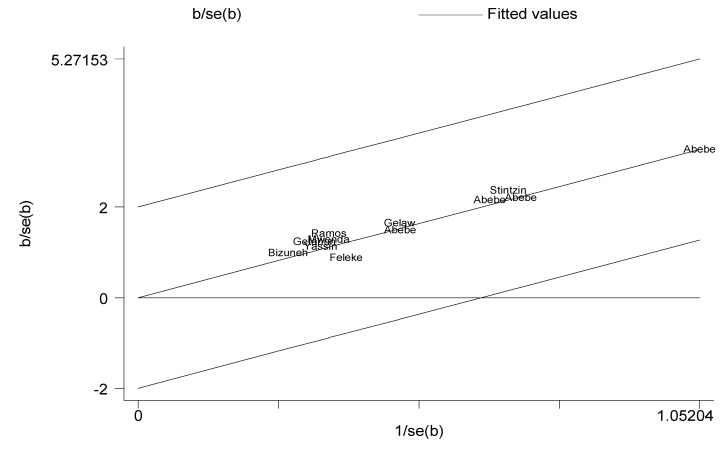
Galbraith plot. Authors’ name represents individual studies. The y-axis represents standard error of the estimates. The x-axis represents the reciprocal of the standard error. In this figure, the weighted estimates of individual studies represented by the authors’ name lie between the ±2 standard error of the pooled estimate. This subjective assessment indicates the absence of heterogeneity between individual studies.

**Figure 3 viruses-12-00062-f003:**
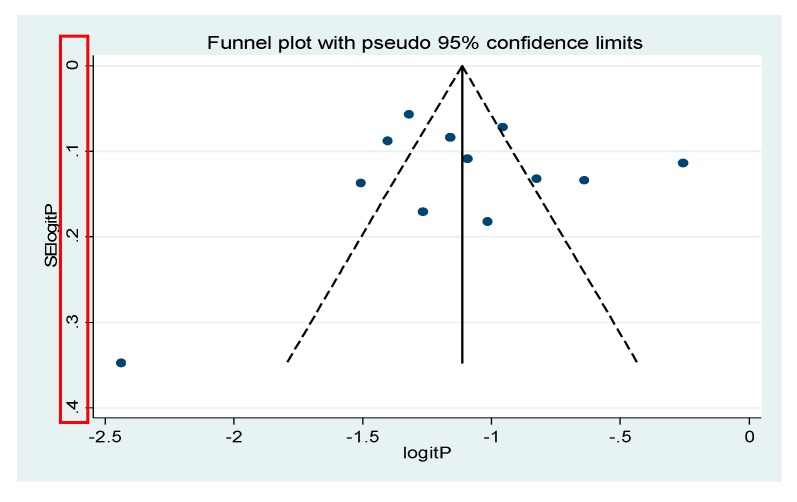
Funnel plot. Each dot represents individual studies. The y-axis represents standard error of estimate. The x-axis represents logit transformed estimates. The subjective assessment of this funnel plot looks asymmetrical which is an indication of heterogeneity of individual studies.

**Figure 4 viruses-12-00062-f004:**
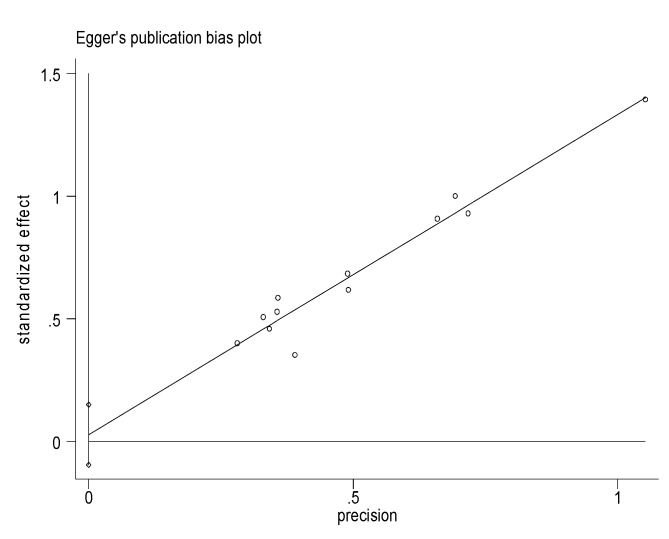
Eggers Publication bias plot. Each dot represents individual studies. The x-axis represents precision (reciprocal of the standard error of the estimate). The y-axis represents standardized effect (estimate divided by its standard error). The subjective assessment was that the y-intercept is closer to the origin (0) which is an indication for the absence of publication bias. This subjective assessment was also supported by the objective Eggers test in the table below which showed the absence of significant publication bias (*p* = 0.619) (Table 2).

**Figure 5 viruses-12-00062-f005:**
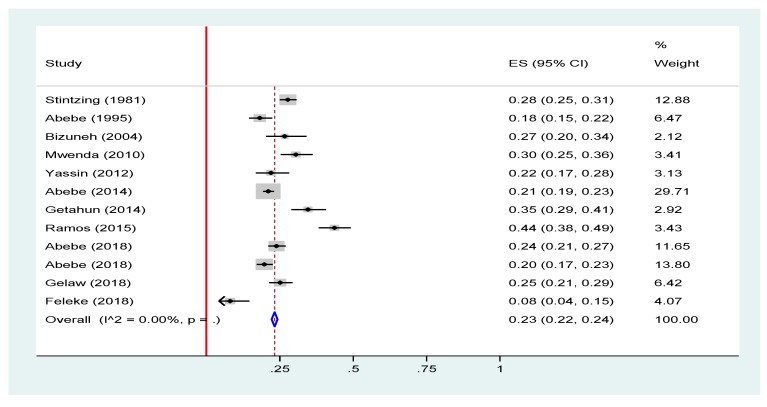
Forest plot of pooled prevalence estimates of rotavirus infection among under-five children with acute gastroenteritis in Ethiopia, 2019 [16,17,18,19,20,21,22,23,24,25,26]. The red line represents the minimum possible prevalence value (0). The dashed line represents the mean pooled rotavirus prevalence estimate. The black dot at the center of the gray box represents the point prevalence estimate of each study and the line indicates the 95% confidence interval of the estimates. The gray box represents the weight of each study contributing to the pooled prevalence estimate. The blue diamond represents the 95% confidence interval of the pooled rotavirus prevalence estimate.

**Figure 6 viruses-12-00062-f006:**
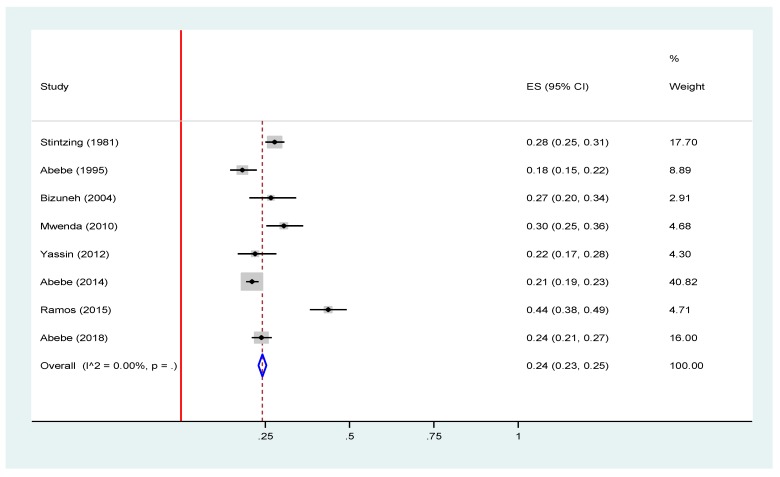
Forest plot of pooled prevalence estimates of rotavirus infection before the introduction of rotavirus vaccine among under-five children with acute gastroenteritis in Ethiopia, 2019 [16,17,19,20,23,24,25,26]. The red line represents the minimum possible prevalence value (0). The dashed line represents the mean pooled rotavirus prevalence estimate. The black dot at the center of the gray box represents the point prevalence estimate of each study and the line indicates the 95% confidence interval of the estimates. The gray box represents the weight of each study contributing to the pooled prevalence estimate. The blue diamond represents the 95% confidence interval of the pooled rotavirus prevalence estimate.

**Figure 7 viruses-12-00062-f007:**
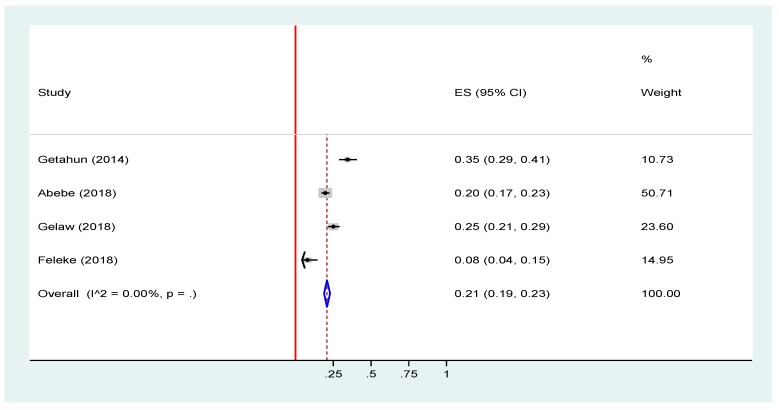
Forest plot of pooled prevalence estimates of rotavirus infection after the introduction of rotavirus vaccine among under-five children with acute gastroenteritis in Ethiopia, 2019 [18,19,21,22]. The red line represents the minimum possible prevalence value (0). The dashed line represents the mean pooled rotavirus prevalence estimate. The black dot at the center of the gray box represents the point prevalence estimate of each study and the line indicates the 95% confidence interval of the estimates. The gray box represents the weight of each study contributing to the pooled prevalence estimate. The blue diamond represents the 95% confidence interval of the pooled rotavirus prevalence estimate.

**Figure 8 viruses-12-00062-f008:**
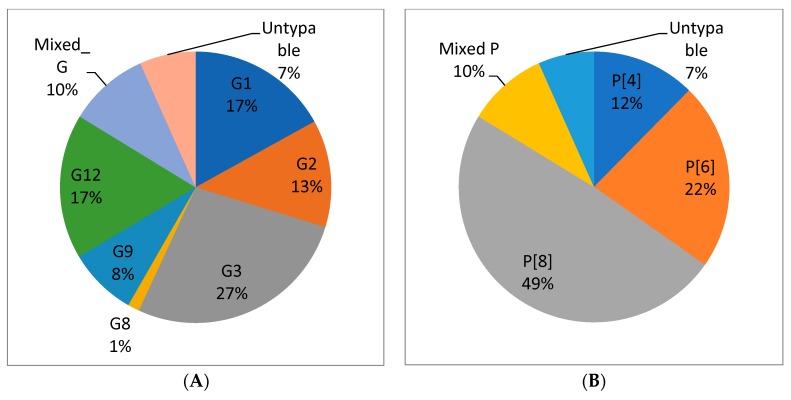
Distribution of rotavirus G types (**A**) and P types (**B**) from children with acute gastroenteritis in Ethiopia from articles published during 2010–2018 [16,19,21,25,26]. *n* = 719 in (**A**), *n* = 719 in (**B**).

**Figure 9 viruses-12-00062-f009:**
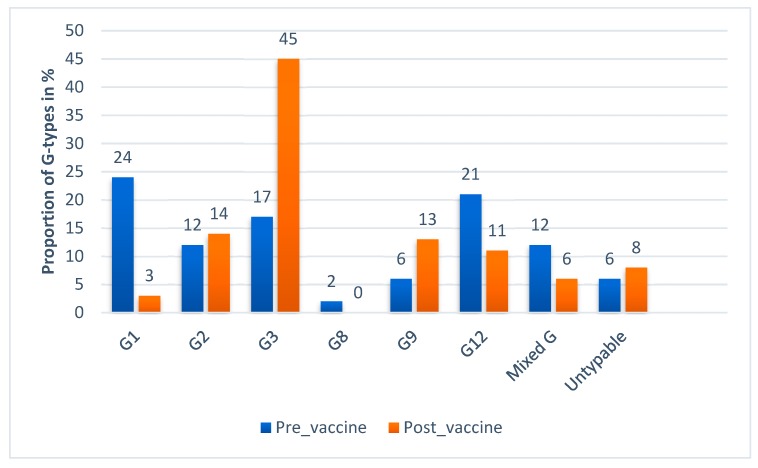
G-type distribution of rotavirus during pre- and post-vaccine introduction in Ethiopia 2010–2018.

**Figure 10 viruses-12-00062-f010:**
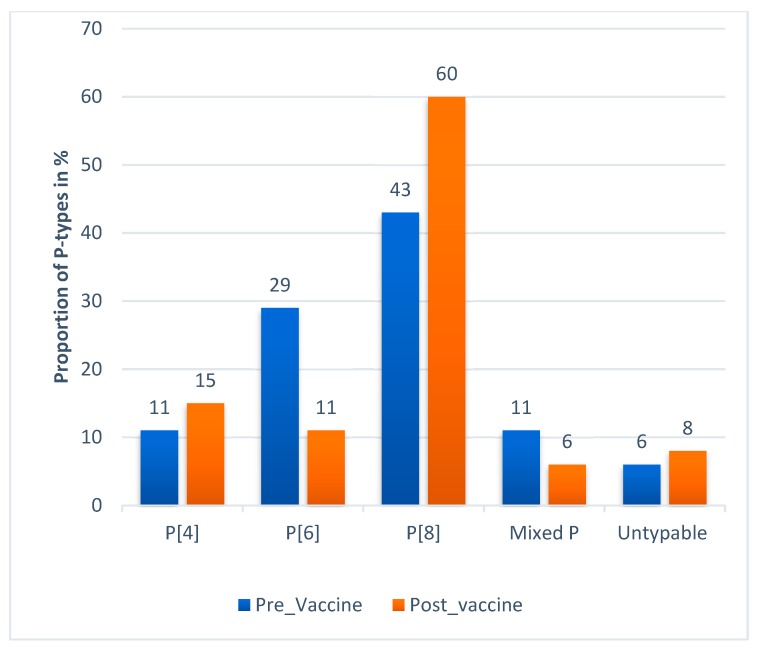
P-type distribution of rotavirus during pre- and post-vaccine introduction in Ethiopia 2010–2018.

**Table 1 viruses-12-00062-t001:** Summary of the studies used in the systematic review and meta-analysis.

No.	Author	Year	Vaccination	Region	Design	Setting	Lab Method	Rotavirus Cases	Sample Size	Proportion (%)	Samples Genotyped	Reference
1	Abebe	2014	Pre-Vaccine	Addis Ababa	Sentinel surveillance	Facility Based	EIA and RT-PCR	388	1841	21.08	215	[16]
2	Stintzing	1981	Pre-Vaccine	Addis Ababa	Cross-sectional	Facility Based	IE	267	962	27.75		[17]
3	Getahun	2014	Post-Vaccine	Addis Ababa	Cross-sectional	Facility Based	EIA	85	246	34.55		[18]
4	Abebe *	2018	Pre-Vaccine	Addis Ababa	Sentinel surveillance	Facility Based	EIA and RT-PCR	188	788	23.86	156	[19]
5	Abebe *	2018	Post-Vaccine	Addis Ababa	Sentinel surveillance	Facility Based	EIA and RT-PCR	161	815	19.75	141	[19]
6	Abebe	1995	Pre-Vaccine	Addis Ababa	Cross-sectional	Facility Based	EIA	65	358	18.16		[20]
7	Gelaw	2018	Post-Vaccine	Amhara	Cross-sectional	Facility Based	RT_PCR	113	450	25.11	113	[21]
8	Feleke	2018	Post-Vaccine	Amhara	Cross-sectional	Community Based	EIA	9	112	8.04		[22]
9	Bizuneh	2004	Pre-Vaccine	Oromia	Cross-sectional	Facility Based	EIA	41	154	26.62		[23]
10	Ramos	2015	Pre-Vaccine	Oromia	Cross-sectional	Facility Based	EIA	137	314	43.63		[24]
11	Yassin	2012	Pre-Vaccine	SNNPR	Cross-sectional	Facility Based	EIA and RT-PCR	44	200	22.00	42	[25]
12	Mwenda	2010	Pre-Vaccine	Addis Ababa	Sentinel surveillance	Facility Based	EIA and RT-PCR	82	269	30.48	52	[26]

EIA = enzyme immune assay, RT-PCR = real time polymerase chain reaction, IE = immunoelectrophoresis, SNNPR = Southern Nations, Nationalities and Peoples Region, * data split into two.

**Table 2 viruses-12-00062-t002:** Egger’s test.

Standard Effect	Coefficient	Standard Error	t	P>|t|	[95% Confidence Interval]
Slope	3.002352	0.2271448	13.22	0.000	2.496242–3.508463
Bias	0.0649642	0.1265539	0.51	0.619	−0.2170154–0.3469438

**Table 3 viruses-12-00062-t003:** G/P combinations of Rotavirus A in Ethiopia, 2012–2018.

G/P Combinations	Pre-Vaccine Introduction Number (%)	Post-Vaccine Introduction Number (%)	Overall Number (%)	References
G1P[4]	1 (0.22)	0 (0)	1 (0.14)	[19]
G1P[6]	21 (4.51)	2 (0.79)	23 (3.2)	[16,19,26]
G1P[8]	92 (19.78)	6 (2.36)	98 (13.6)	[16,19,25,26]
G2P[4]	49 (10.54)	36 (14.17)	85 (11.8)	[16,19,21,25,26]
G2P[6]	7 (1.5)	0 (0)	7 (1.0)	[16]
G3P[6]	78 (16.77)	24 (9.45)	102 (14.2)	[16,19,21,25,26]
G3P[8]	2 (0.43)	91 (35.83)	93 (12.9)	[19,21]
G8P[6]	9 (1.94)	0 (0)	9 (1.3)	[25,26]
G8P[8]	1 (0.22)	0 (0)	1 (0.14)	[26]
G9P[4]	0 (0)	1 (0.39)	1 (0.14)	[19]
G9P[6]	9 (1.94)	0 (0)	9 (1.3)	[16,19]
G9P[8]	18 (3.87)	31 (12.2)	49 (6.8)	[16,19,21,25]
G12P[4]	0 (0)	2 (0.79)	2 (0.27)	[19]
G12P[6]	10 (2.15)	1 (0.39)	11 (1.5)	[16,19,25]
G12P[8]	86 (18.49)	25 (9.94)	111 (15.4)	[16,19,21,25,26]
Mixed-G/P	54 (11.61)	15 (3.91)	69 (9.6)	[16,19,26]
Untypable	28 (6.02)	20 (7.87)	48 (6.7)	[16,19,26]
Total	465 (100)	254 (100)	719 (100)

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
