# Peer review of "Prevalence and Genetic Diversity of Rotaviruses among under-Five Children in Ethiopia: A Systematic Review and Meta-Analysis"

_viruses, 2020, doi:10.3390/v12010062_

Round 1

Reviewer 1 Report

Prevalence and genetic diversity of rotaviruses among under-five children in Ethiopia: a systematic review and meta-analysis

By Debasu Damtie Gella* et al (*Corresponding authors: DDG, Anastasia N Vlasova) Submitted to Viruses (Editorial No. viruses-674568) 

Note to the Editor

This is a useful review of the prevalence of species A rotaviruses (RVA) and their genotype variability in under 5  year-old children with acute gastroenteritis in Ethiopia during the years… [authors, please insert]. Apart from minor clarifications a more transparent presentation and explanation of the statistical evaluation is requested.

Accordingly, a moderate revision of the manuscript is recommended. Once a revised manuscript has been received which is to the satisfaction of the Editor, it should be published. [The revised manuscript does not have to be seen by this reviewer again.]

Specific Comments

Page

1          Title. Consider rephrasing: ‘Prevalence and genetic diversity of rotaviruses among <5 year-old children with acute gastroenteritis in Ethiopia during…  [please insert time period]: review and meta-analysis’, or similar.

            Abstract, line 1. Consider reading: ‘Rotavirus infections are a major cause…’. Line 3. … among children under five years of age and… Penultimate line: … period. Further studies…

           Line 6 from bottom. … globally after respiratory tract infections…

            Line 4 from bottom. Ref [3] is slightly outdated. Consider citation of a more recent review.

2          Line 1. … double-stranded RNA and form a genus of the Reoviridae family. The viral genome encodes…

            Line 6. Consider phrasing: … via point mutations, gene reassortment, gene recombination and zoonotic transmission… Consider citation of: Desselberger U. Rotaviruses. Virus Res. 2014; 190: 76-95.

Paragraph 2, line 7. Ref [14] lists other reasons for variation in efficacy/effectiveness of rotavirus vaccines in low-income countries.

Paragraph 3. … CoCoPo approach. Please provide a reference.  … under 5 year-old children… [throughout manuscript].

3          Paragraph 3, line 8. … Reverse Transcription-Polymerase Chain Reaction…  The studies were disaggregated for pre- and post-vaccination periods… (RT-PCR)…

            Paragraph 4. … Tables for data extraction were developed…

            Line 3 from bottom. … of genotyped samples. Heterogeneity tests were carried out by… Here some clarification and citation of appropriate refs are required.

6          Table 1. Ref. no. 3 should have ‘Mekonen’ as author.

            The footnote should read: … RT-PCR = Reverse Transcription-Polymerase Chain Reaction…

7          Fig. 2 requires a footnote explaining what is shown.

8          Figs. 3 and 4 require footnotes explaining what is shown.

            Table 2 requires a footnote explaining what is shown.

9          Fig. 5. requires a footnote explaining what is shown. The significance of the I2 value should be explained. The numbers of the ref. list should be added to the ‘study’ column. Reference ‘Getahun, 2014’ is not found in the ref. list.

10        Fig. 6. It should be considered to transform the Fig. data into a Table; this would make the comparison with Table 3 easier.

            End of Results and Table 3. The high values of infections with mixed or untypable genotypes should be commented on. The following refs could be considered for citation:

            Ahmed, 1991

            Israeli study, 2….

            Line 6 from bottom. … is caused by rotavirus A infection and is consistent with…

11        Paragraph 2, line 7. Consider phrasing: … The genotype G8 which is unusual in humans was found in a few isolates; it is very common in cattle, suggesting that …

            Line 3 from bottom. Ref [14] is not relevant for this argument. The G1P[8] rotavirus vaccine (Rotarix) is cross-protective. See refs.

            NEnglj Med 2006

            Vesikari et al, Lancet 2007;

12        Ref [5]. Read: Estes MK, Greenberg HB. Rotaviruses …

            Ref [7]. Check authors.

13        Ref [15]… National Expanded Programme of Immunization: Comprehensive multi-year plan…  

Author Response

Comments and Suggestions for Authors

RE1: The authors performed extensive literature review on prevalence of rotavirus infection among acute gastroenteritis in Ethiopia.  Given the current difficulty to improve rotavirus efficacy in low income countries results from this study could provide useful information in the status of rotavirus infection over time.  However, there are some issues the authors need to address.

AU: We thank the reviewer for the positive comments and constructive suggestions.

RE: The authors stated that part of the objectives of the review was to study if rotavirus serotype distribution changed as result of vaccination in Ethiopia, however, the analysis did not appear in the paper. Authors please explain.

AU: The objective of the review was actually to determine the pooled prevalence and genotype distribution of rotavirus induced acute gastroenteritis in children under 5 and to see if those were different from other geographic regions and if they included strains of zoonotic origin. As there was no heterogeneity in prevalencebetween the primary studies, we didn’t consider analyzing the data as pre-vaccine and post-vaccine introduction separately. Additionally, there were only two studies reporting genotype data in post-vaccine period, which is insufficient for a comprehensive overview of the genotype dynamics associated with vaccination. However, we found the comment important and included the analysis for pre- and post-vaccine introduction data separately for ease comparison of the results. Please have a look at the changes on page 9 (paragraph 1 and Figure 6), page 10 (Figure 7), page 11 (paragraphs 1&2), page 12 (Figure 9&10) and page 13 (paragraph 1 and Table 3).

RE: The authors presented multiple graphs and table to analyze heterogeneity of the data. For lay persons who may not be familiar with these analyses authors please give more explanations how to interpret the data.  What the heterogeneity means and what is its effect for overall analyses. 

AU: Thank you very much for the comment. We have included some information about the what heterogeneity means on page 7 paragraph 1 in addition to the existing information on methods (statistical analysis section) on page 3 last paragraph. Moreover, legends are included for figure 2, figure 3 and figure 4 to ease data interpretation.

RE:  Is the prevalence rate of one community-based study (8.04%) within the range of other facility-based studies? 

AU: Yes. As it can be seen from the forest plot (Figure 5) on page 9, the 95% CI estimate ranges from 4%-15% having low precision associated with smaller sample size. The upper limit of this study overlaps with the lower limit of other study (Abebe 1995). Therefore, there is no statistically significant difference between these two studies.

RE: Figure 6 and table 3, it is not clear if authors observed changes in rotavirus serotype distribution before or after implementation of vaccination. Authors please explain and discuss.

AU: It is true that presenting the rotavirus serotype distribution data before and after introduction of rotavirus vaccine is important to see the effect of the vaccine on the prevalence of rotavirus induced acute gastroenteritis and genotypic distributions of rotavirus. We have included the data accordingly. Please have a look at the changes on page 9 (paragraph 1 and Figure 6), page 10 (Figure 7), page 11 (paragraphs 1&2), page 12 (Figure 9&10) and page 13 (paragraph 1 and Table 3).

RE: In fourth paragraph of the discussion, “This finding is comparable with a meta-analysis study done in Africa which reported G1P[8] (22.64%...”. The G/P combination (G12P[8], G3P[3], etc) in this paper is obviously different from other Africa study.  The authors please clarify.

AU: We thank the reviewer for the comment and we revised our writing to accurately reflect what was similar between ours and previous findings and what differed.

RE: Can the authors correlate that appearance of G12 rotaviruses in Ethiopia with the implementation of vaccination? Did they observe changes in prevalence rate or rotavirus serotype distribution pre- or post-vaccination?  Please discuss. 

AU: Although, it was noted previously that G12 genotype became one of the major emerging genotypes in the post-vaccine introduction period, our analysis indicated that the distribution of G12 post vaccine introduction seems decreased compared to pre-vaccine introduction in this study. However, the limited amount of data available for the analysis (only 2 manuscripts) may be insufficient to definitively establish if G12 emergence in Ethiopia was associated with vaccination. Overall, the role of rotavirus vaccine for selective pressure of other rotavirus genotypes would result in emergence of new genotypes such as G12 genotypes as indicated by other studies cited in the manuscript.

RE: In selection of criteria of study: number III stated “the number of study subjects with positive and negative result must be reported. Did the authors mean rotavirus positive or negative results?

AU: We thank the reviewer, we meant by rotavirus negative or positive results. The criterium is amended as per the reviewer’s comment. Please have a look at on page 3.  

RE: Authors stated that five studies used RT-PCR as genotyping method but only one study in table 1 appeared using RT-PCR. Please clarify.

AU: We thank the reviewer for the comment. What we showed in the table is the detection method used for the determination of rotavirus prevalence. Only one study used RT-PCR for the determination of the prevalence of rotavirus infection while others used EIA. However, all studies involving genotyping have used RT-PCR in addition to EIA. We made some amendments to the table for clarity. Please have a look at table 1 on page 6.

Reviewer 2 Report

The authors performed extensive literature review on prevalence of rotavirus infection among acute gastroenteritis in Ethiopia.  Given the current difficulty to improve rotavirus efficacy in low income countries results from this study could provide useful information in the status of rotavirus infection over time.  However, there are some issues the authors need to address.

The authors stated that part of the objectives of the review was to study if rotavirus serotype distribution changed as result of vaccination in Ethiopia, however, the analysis did not appear in the paper. Authors please explain. The authors presented multiple graphs and table to analyze heterogeneity of the data. For lay persons who may not be familiar with these analyses authors please give more explanations how to interpret the data.  What the heterogeneity means and what is its effect for overall analyses.   Is the prevalence rate of one community based study (8.04%) within the range of other facility based studies?  Figure 6 and table 3, it is not clear if authors observed changes in rotavirus serotype distribution before or after implementation of vaccination. Authors please explain and discuss. In fourth paragraph of the discussion, “This finding is comparable with a meta-analysis study done in Africa which reported G1P[8] (22.64%...”. The G/P combination (G12P[8], G3P[3], etc) in this paper is obviously different from other Africa study.  The authors please clarify. Can the authors correlate that appearance of G12 rotaviruses in Ethiopia with the implementation of vaccination? Did they observe changes in prevalence rate or rotavirus serotype distribution pre- or post-vaccination?  Please discuss.  In selection of criteria of study: number III stated “the number of study subjects with positive and negative result must be reported. Did the authors mean rotavirus positive or negative results? Authors stated that five studies used RT-PCR as genotyping method but only one study in table 1 appeared using RT-PCR. Please clarify.

Author Response

Comments and Suggestions for Authors

RE2: While rotavirus infection is a major cause of acute gastroenteritis in children worldwide, the sub-Saharan region of Africa is disproportionately affected by the disease. This review and meta-analysis by Gella et al. pooled literature to investigate the pooled genotype prevalence of rotaviruses infecting children under the age of five in Ethiopia. This study found that the dominant rotavirus strains in the region were G12P[8] (15.43%), G3P[6] (14.2%), G1P[8] (13.6%) and G3P[8] (12.9%). Critically, the authors note that the current Rotarix vaccine available in the region only covers the G1P[8] strain, raising the need for the development of new strain specific vaccines.

AU: we thank the reviewer for the positive comments and constructive suggestions.

 Major comments:

RE: The figure legends are unacceptably lacking in description, at times only stating the type of graph used. Please rewrite figure legends in this text to comprehensively address the analyses used in this study.

AU: Thank you for the comment. As per the reviewer’s comment, legends are included for figure 2, figure 3 and figure 4 for ease interpretation. Please have a look at the figure legends on page 7 & 8.

Minor comments:

RE: Additionally, the dimensions of some figures are causing formatting issues for some of the accompanying text (refer to Table 1 and Figure 6 for examples). These should be adjusted for better legibility.

AU: The comment of the reviewer is well taken and adjustments are made to Table 1 and figure 6 (now Figure 8) on page 6 and 11 respectively.

Reviewer 3 Report

While rotavirus infection is a major cause of acute gastroenteritis in children worldwide, the sub-Saharan region of Africa is disproportionately affected by the disease. This review and meta-analysis by Gella et al. pooled literature to investigate the pooled genotype prevalence of rotaviruses infecting children under the age of five in Ethiopia. This study found that the dominant rotavirus strains in the region were G12P[8] (15.43%), G3P[6] (14.2%), G1P[8] (13.6%) and G3P[8] (12.9%). Critically, the authors note that the current Rotarix vaccine available in the region only covers the G1P[8] strain, raising the need for the development of new strain specific vaccines.

Major comments:

 The figure legends are unacceptably lacking in description, at times only stating the type of graph used. Please rewrite figure legends in this text to comprehensively address the analyses used in this study.

Minor comments:

Additionally, the dimensions of some figures are causing formatting issues for some of the accompanying text (refer to Table 1 and Figure 6 for examples). These should be adjusted for better legibility.

Author Response

RE: This systematic review presents a meta-analysis of 11 different epidemiological studies of rotavirus-induced disease in children under five years old in Ethiopia. This is a valuable review that determines the overall prevalence of rotavirus in children with gastroenteritis and considers the range of genotypes circulating.

AU: We thank the reviver for the positive comments and constructive suggestions.

RE: The methods used to select the papers studied are robust, and details of these are presented clearly.

AU: We thank the reviewer again for the positive comment.

RE: The main query I have for this manuscript is regarding the use of vaccination. All selected studies are separated into pre and post-vaccination (section 3.4), and this is clearly highlighted in table 1. However, all results then focus on prevalence and genotypes across all years, with no pre and post 2013 comparison. Is it possible to perform analyses on these two separate groups? It would be really interesting to see if there are any differences. Alternatively, it needs to be explained why this isn’t performed.

AU: It is true that presenting the pooled pre-vaccine and post-vaccine introduction prevalence data is important to see the effect of the vaccine on the prevalence of rotavirus induced acute gastroenteritis. However, during heterogeneity analysis there was no statistically significant difference among studies across all years. As a result, we pooled the overall pooled prevalence of rotavirus infection without pre versus post vaccine dichotomy. However, we found the comment important and included the analysis for pre and post vaccine introduction data separately for ease comparison of the results. Please have a look at the changes on page 9 (paragraph 1 and Figure 6), page 10 (Figure 7), page 11 (paragraphs 1&2), page 12 (Figure 9&10) and page 13 (paragraph 1 and Table 3).

RE: Regarding the effect of rotavirus vaccine uptake rates, it would be useful to consider the recent publication by Wondimu et al 2019 that examines this in Ethiopia in 2016. Uptake was shown it to be approximately 56% (I appreciate this paper will not have been published at the time this review was performed). Given the low efficacy rates of the rotavirus vaccine in other African countries, it would be valuable to see if this estimated modest number (~25%?) of protected children would be impacting upon prevalence and genotype distribution.

AU: Although our objective of this review was not to see the impact of vaccination on the prevalence of rotavirus infection, we have now analyzed the data in in pre- post-vaccine periods independently and there was no significant difference in the prevalence of rotavirus. However, the genotype distribution seems different in pre- verses post-vaccine periods. All the data is included in the manuscript on page 10-13.

 Minor comments are as follows:

RE: Introduction: Page 2: Listing of P genotypes that represent 90% strains worldwide: need brackets around P[6]

AU: We thank you for the comment; it is corrected.

RE: Table 1: Study 5 – need samples genotyped (as study 4 and 5 are part of the same report, I’m assuming there are genotypes samples from both the pre and post vaccine periods?)

AU: We thank the reviewer. We have separated the data as per and post vaccine period. Please have a look at Table 1 on page 6.

RE: I am unable to comment on figures 2-4 due to lack of familiarity with these approaches to data analysis, but the forest plot in figure 5 is really clear and provides a good summary of the results.

AU: Thank you for the reflection. We have added legends to the figures for better understanding.

RE: Discussion: The pan-African systematic review cited identified 40% prevalence in rotavirus diarrhoea which was substantially different to the results of this study. Differences in inclusion criteria were noted, but as this study was published in 2010, do the authors think that vaccination might also being having an impact? Similarly, the other systematic review cited from Latin American and the Caribbean, which had a very similar prevalence rate was published in the pre-vaccine era (2011), so cannot be directly comparable assuming the rotavirus vaccine is moderately effective

AU: We thank the reviewer for the comment. Of course, the difference could be due to the introduction of vaccine in addition to the inclusion criteria we already indicated. We have modified the statement considering the reviewer’s comment. Please have a look at on page 14.   

Reviewer 4 Report

This systematic review presents a meta-analysis of 11 different epidemiological studies of rotavirus-induced disease in children under five years old in Ethiopia. This is a valuable review that determines the overall prevalence of rotavirus in children with gastroenteritis and considers the range of genotypes circulating.

The methods used to select the papers studied are robust, and details of these are presented clearly.

The main query I have for this manuscript is regarding the use of vaccination. All selected studies are separated into pre and post-vaccination (section 3.4), and this is clearly highlighted in table 1. However, all results then focus on prevalence and genotypes across all years, with no pre and post 2013 comparison. Is it possible to perform analyses on these two separate groups? It would be really interesting to see if there are any differences. Alternatively, it needs to be explained why this isn’t performed.

Regarding the effect of rotavirus vaccine uptake rates, it would be useful to consider the recent publication by Wondimu et al 2019 that examines this in Ethiopia in 2016. Uptake was shown it to be approximately 56% (I appreciate this paper will not have been published at the time this review was performed). Given the low efficacy rates of the rotavirus vaccine in other African countries, it would be valuable to see if this estimated modest number (~25%?) of protected children would be impacting upon prevalence and genotype distribution.

Minor comments are as follows:

Introduction: Page 2: Listing of P genotypes that represent 90% strains worldwide: need brackets around P[6]

Table 1: Study 5 – need samples genotyped (as study 4 and 5 are part of the same report, I’m assuming there are genotypes samples from both the pre and post vaccine periods?)

I am unable to comment on figures 2-4 due to lack of familiarity with these approaches to data analysis, but the forest plot in figure 5 is really clear and provides a good summary of the results.

Discussion: The pan-African systematic review cited identified 40% prevalence in rotavirus diarrhoea which was substantially different to the results of this study. Differences in inclusion criteria were noted, but as this study was published in 2010, do the authors think that vaccination might also being having an impact? Similarly, the other systematic review cited from Latin American and the Caribbean, which had a very similar prevalence rate was published in the pre-vaccine era (2011), so cannot be directly comparable assuming the rotavirus vaccine is moderately effective

Author Response

RE: This systematic review presents a meta-analysis of 11 different epidemiological studies of rotavirus-induced disease in children under five years old in Ethiopia. This is a valuable review that determines the overall prevalence of rotavirus in children with gastroenteritis and considers the range of genotypes circulating.

AU: We thank the reviver for the positive comments and constructive suggestions.

RE: The methods used to select the papers studied are robust, and details of these are presented clearly.

AU: We thank the reviewer again for the positive comment.

RE: The main query I have for this manuscript is regarding the use of vaccination. All selected studies are separated into pre and post-vaccination (section 3.4), and this is clearly highlighted in table 1. However, all results then focus on prevalence and genotypes across all years, with no pre and post 2013 comparison. Is it possible to perform analyses on these two separate groups? It would be really interesting to see if there are any differences. Alternatively, it needs to be explained why this isn’t performed.

AU: It is true that presenting the pooled pre-vaccine and post-vaccine introduction prevalence data is important to see the effect of the vaccine on the prevalence of rotavirus induced acute gastroenteritis. However, during heterogeneity analysis there was no statistically significant difference among studies across all years. As a result, we pooled the overall pooled prevalence of rotavirus infection without pre versus post vaccine dichotomy. However, we found the comment important and included the analysis for pre and post vaccine introduction data separately for ease comparison of the results. Please have a look at the changes on page 9 (paragraph 1 and Figure 6), page 10 (Figure 7), page 11 (paragraphs 1&2), page 12 (Figure 9&10) and page 13 (paragraph 1 and Table 3).

RE: Regarding the effect of rotavirus vaccine uptake rates, it would be useful to consider the recent publication by Wondimu et al 2019 that examines this in Ethiopia in 2016. Uptake was shown it to be approximately 56% (I appreciate this paper will not have been published at the time this review was performed). Given the low efficacy rates of the rotavirus vaccine in other African countries, it would be valuable to see if this estimated modest number (~25%?) of protected children would be impacting upon prevalence and genotype distribution.

AU: Although our objective of this review was not to see the impact of vaccination on the prevalence of rotavirus infection, we have now analyzed the data in in pre- post-vaccine periods independently and there was no significant difference in the prevalence of rotavirus. However, the genotype distribution seems different in pre- verses post-vaccine periods. All the data is included in the manuscript on page 10-13.